# Using Lean Six Sigma in a Private Hospital Setting to Reduce Trauma Orthopedic Patient Waiting Times and Associated Administrative and Consultant Caseload

**DOI:** 10.3390/healthcare11192626

**Published:** 2023-09-26

**Authors:** Anthony Pierce, Seán Paul Teeling, Martin McNamara, Brendan O’Daly, Ailish Daly

**Affiliations:** 1Beacon Hospital, Beacon Court, Bracken Rd, Sandyford Business Park, Sandyford, D18 AK68 Dublin, Ireland; 2UCD Centre for Interdisciplinary Research, Education & Innovation in Health Systems, School of Nursing, Midwifery & Health Systems UCD Health Sciences Centre, D04 VIW8 Dublin, Ireland; sean.p.teeling@ucd.ie (S.P.T.);; 3Centre for Person-Centered Practice Research Division of Nursing, School of Health Sciences, Queen Margaret University Drive, Queen Margaret University, Musselburgh EH21 6UU, UK

**Keywords:** wait time, trauma orthopedics, triage, Gemba, Voice of Customer, Lean Six Sigma

## Abstract

In Ireland, the extent of outpatient orthopedic waiting lists results in long waiting times for patients, delays in processing referrals, and variation in the consultant caseload. At the study site, the Define, Measure, Analyze, Improve, and Control (DMAIC) Lean Six Sigma framework was applied to evaluate sources of Non-Value-Added (NVA) activity in the process of registering and triaging patients referred to the trauma orthopedic service from the Emergency Department. A pre- (October–December 2021)/post- (April–August 2022) intervention design was employed, utilizing Gemba, Process Mapping, and the TIMWOODS tool. Embracing a person-centered approach, stakeholder Voice of Customer feedback was sought at each stage of the improvement process. Following data collection and analysis, a co-designed pilot intervention (March 2022) was implemented, consisting of a new triage template, dedicated trauma clinic slots, a consultant triage roster, and a new option to refer directly to physiotherapy services. This resulted in the total wait time of patients for review being reduced by 34%, a 51% reduction in the process steps required for registering, and an increase in orthopedic consultant clinic capacity of 22%. The reduction in NVA activities in the process and the increase in management options for triaging consultants have delivered a more efficient trauma and orthopedic pathway.

## 1. Introduction

In Ireland, musculoskeletal injuries are the most common reason for the presentation of patients to the Emergency Department (ED) [1]. The volume of musculoskeletal presentations to the ED, in turn, leads to a large number of referrals to trauma orthopedic services [1]. These injuries represent one-third of bed days used in hospitals and one-third of the reasons for acute surgery in Ireland each year [2]. In 2015, orthopedics was reported to have the highest volume of outpatient activity in the Irish health service [2], and recent analysis indicates a further 15% increase [3], with 10,393 patients on orthopedic waiting lists in April 2021 [4]. Similarly, high levels of outpatient activity and orthopedic surgery waiting times have been reported internationally [5,6]. Worldwide, the increase in waiting time for orthopedic and musculoskeletal complaints has negatively affected patient satisfaction [7,8].

This study was carried out in a private hospital in South Dublin, Ireland, and addresses similar processes to previous Lean Six Sigma (LSS) studies completed at the same site relating to patient flow from the ED [9] and medical documentation [10]. The hospital’s ED has over 13,000 presentations each year, with an average of 10% being referred to the trauma orthopedic service. In early 2022, overall presentations to the ED increased by 11% compared to the previous year, with a concurrent rise in ED referrals to trauma orthopedics. These referrals are triaged by consultants who assess the degree of case severity based on presenting signs and symptoms and assign an order of clinical priority [1]. The increasing volume heightens the responsibility of the six trauma orthopedic consultants who facilitate the pathway in combination with their own elective orthopedic caseloads and Public Health Service Executive (HSE) work. Many referrals to the orthopedic service do not require a review appointment with a consultant doctor [11] to continue accessing appropriate care. Unnecessary consultant appointments generate an associated volume of work, further increasing pressure on an already busy service [3].

As the study site has a large number of experienced staff trained in LSS (*n* = 382) and a track record of successful LSS improvement, it was the methodology of choice for this study. LSS is a merge of two process improvement methodologies: Lean, developed by Toyota, and Six Sigma, developed by Motorola [12]. Lean aims to eliminate waste or NVA activity, often using qualitative measures to achieve this goal [13], whereas Six Sigma targets variation and employs statistical analysis [14]. Additionally, both methodologies have contributed positively to solving problems when combined with a person-centered improvement approach [12] in ED and outpatient settings [15]. Therefore, our work was informed by a combined person-centered Lean Six Sigma approach [12].

A review of 66 articles (year range 2001–2018) relevant to LSS deployment in health systems reported that the methodology is effective in reducing overall waiting times for public hospital appointments [16]. Although there are no specific studies reviewing waiting lists (measured in days) for an outpatient orthopedic department, a reduction of 18% in median outpatient ophthalmology waiting times was achieved using LSS [17]. Within inpatient orthopedics, organizations have applied LSS tools to target length of stay (LOS) [18] or to improve disease-specific pathways [19] such as a reduction in LOS by 42% in 148 knee replacement patients [20]. Improta and colleagues [20] noted NVA in pre-operative planning and a lack of standardized procedures.

The main goal of this study was to reduce the average wait time for patients referred from the ED to trauma orthopedics. At the study site, previous LSS studies had identified duplication of administrative work in the registration of elective orthopedic surgeries [21]. Therefore, in addition to targeting patient waiting times, the project evaluated the impact of the reduction in wait times on the administrative process and consultant clinic caseload. The hypothesis was that improvements in these areas would improve satisfaction for both patients and staff.

## 2. Methods

LSS methodology was applied using the DMAIC improvement framework. The letters in the acronym stand for Define, Measure, Analyze, Improve, and Control. This framework structures the quality improvement process, encourages stakeholder engagement, and facilitates an implementation plan for devised solutions [16]. LSS methodology was employed via a pre- and post-intervention study design to measure variables relating to wait times, patient registration, and clinical caseload [21]. The methods used in the study are now outlined using the structure of the DMAIC framework.

### 2.1. Define Phase

A multidisciplinary project team was established to deliver the process improvement project. Membership included the lead trauma orthopedic surgeon, the orthopedic patient services lead, and the orthopedic patient services team. A project charter, one of a number of LSS tools (Table 1) used to structure the study, was completed to define the problem, document the project goal, and consider any risks associated with implementation [22]. Due to the large number of referrals and to enable a solution-focused proof of concept, the scope of the study was refined to include only ED patients referred for peripheral musculoskeletal symptoms such as proximal humerus fractures, distal fibular fractures, and ankle sprains. The scope excluded patients where a care pathway already existed, e.g., hand, spinal, or post-operative patients. An initial request for feedback from staff directly involved in the process determined the current process to be a source of staff ‘anxiety’, to be ‘time-consuming’, and identified that ‘certain referrals generally do not need to see a trauma surgeon’.

To identify what quantitative data existed to capture the areas of NVA in the process, we completed an initial analysis of wait time data over a four-week period (20 September–22 October 2021). On the evaluation of this baseline data, a mean wait time for patients of 15.98 days was noted. An interquartile range of 15 days reflected the variability in the process wait time. The qualitative and quantitative baseline data were used to formulate a Critical to Quality (CTQ) tool. This is a tool that enables the information gathered from customers to be developed into critical process requirements that are measurable (Table 1) [25]. After our initial Voice of Customer (VOC) gathering and the completion of a CTQ, the key metrics for data collection were identified as:Waiting times:
for a patient to be registered in the orthopedic service as a trauma orthopedic referral from the ED and informed of the receipt of their referral via text message;for the consultant to triage the referral;for the patient to attend their orthopedic trauma appointment following the ED referral.Time taken and number of steps or touch points (points at which staff come into contact with the patient or action the referral) to:
register a referral;triage a referral;book the patient’s appointment.The efficiency of the trauma consultant’s (total *n* = 6) caseload is represented by the following:
opportunity to review the elective caseload;conversion rate from appointment to surgery.

### 2.2. Measure Phase

To support our data collection, we completed a Gemba walk. This is a technique where the project team goes to the actual place of work, examines the process steps in real-time, and records the observations [27]. From this, we were able to formulate a high-level process map (Figure 1) [29]. This gave the project team an overview of the patient’s journey from ED referral to orthopedic surgeon consultation. The project scope centered on the final four steps (steps 12–15), which included the registration, triage, and booking processes for patients referred to the orthopedic service from the ED.

Gemba walks were timed to gather quantitative data on the duration of each step in the orthopedic patient services registration process. The number of touch points required to complete the registration and triage processes was also recorded. A touch point was defined as any physical interaction by orthopedic patient services or the trauma consultant with an IT system in order to progress the referral. The number of times staff had to switch between different hospital IT systems to complete this process was also recorded. Observations from the Gemba walk are illustrated on the process map (Figure 1), which is a graphical representation of the process [29]. This LSS tool facilitates visualization of the entire process, establishes process limitations, and addresses the question, ‘what the process actually is?’. The process map was developed in December 2021 following repeated Gemba and extensive VOC engagement. To reduce the risk of researcher misinterpretation of data, collated numerical information and observations were directly corroborated by the members of ED, patient services, and consultants who validated the map. A VOC was completed with the patient services team directly involved in registering ED referrals (*n* = 4) and with three trauma orthopedic consultants.

Patients routinely arrived at the orthopedic center only on the day of their appointment. Their wait time, the triage time, and the time taken (in days) to book their appointment were recorded by means of a retrospective audit of data from the hospital’s existing information system, MEDITECH, over three months (October–December 2021). The results of this audit are visualized as a linear process map (Figure 2) of pre-intervention or baseline waiting times for each phase of the patient journey, including registration, triage decision, and total appointment wait. The map also shows the baseline outcomes relative to the hospital’s key performance indicators (KPIs).

To understand the patient’s perspective of the trauma orthopedic service, a purposive sample of all patients (*n* = 189) who attended ED over this pre-intervention period (October–December 2021) was surveyed by phone call (*n* = 30). Patients who attended the trauma orthopedic consultation were assigned a number, which was entered into a random number generator. Patients randomly selected were subsequently contacted. Ten patients per month were surveyed in an attempt to reduce temporal bias. The composition of consultant clinic caseload and surgeries was also gathered from MEDITECH over a five-month period (January–May 2022). The data collected from the orthopedic patient services questionnaires, consultant surveys, and patient surveys gave the project team an understanding of the NVA from the viewpoint of the patient, the staff, and the organization and the potential impact this had on satisfaction.

### 2.3. Analyze Phase

To evaluate the NVA in the data gathered from the VOC, Gemba, and audit, a TIMWOODS tool was used (Table 2). Each word of the acronym represents one of eight potential wastes in a process [30]. The VOC illustrated that patient services staff felt excessive time was required to complete the registration tasks; there were too many opportunities for manual error and the consultants’ belief that certain presentations could be managed differently. These themes representing waste were then evaluated further with stakeholders. This involved using an Ishikawa Fishbone diagram to enable a root cause analysis of the NVA identified and to target areas for improvement [27,28].

### 2.4. Improve Phase

Working collaboratively with stakeholders, we consolidated our analysis of NVA and root causes to enable us to co-design potential solutions that included:(1)Designing an IT system to track and triage patients (the “Patient Services Homework Tracker”).(2)Ring-fencing trauma orthopedic slots.(3)Developing an information leaflet about the pathway for staff and patients.(4)Offering physiotherapy as opposed to consultant review (where clinically indicated) as an option for patients.(5)Creating a consultant triage roster to enable optimal communication between the team with improved visibility of staffing.

A PICK (Possible, Implement, Challenge, Kill) chart was used to prioritize the potential improvements with the greatest positive yield [32]. The project team and stakeholders identified and categorized ideas according to how easy they would be to implement and their likely impact. A practicality tool [30] was subsequently used to consider and rank which of these solutions would be realistic for this particular setting.

The ‘Patient Services Homework Tracker’ (solution 1) was rated as the solution likely to have the greatest beneficial impact. A patient tracker was already used in the ED, following patients’ progress from registration to diagnostics and discharge. The ED Tracker was used as a method of observing patient progress in the department, including wait times and the onward referral destination. However, the ED Tracker stopped monitoring the patient’s journey at ED discharge and did not facilitate patient triage.

To address our problem, the existing tracker required adaptation to enable continued tracking of patients’ progress after ED discharge to orthopedic outpatient registration. In addition, the team suggested the tracker should include the opportunity for orthopedic consultants to triage patients electronically when they are not on the hospital campus. This would permit a quicker triage decision as many of the consultants had clinics operating at other hospital sites each week. Additionally, the triaging consultant would have the option to advise the patient to attend physiotherapy prior to seeing a consultant, if appropriate. This would also reduce patient follow-up wait times. Another suggested benefit of the tracker was improved communication between the emergency and orthopedic departments. The risk of error was also reduced as the current process of using a simple Excel sheet to log referrals would be discontinued.

The ‘Patient Services Homework Tracker’ was co-designed with patient services, IT, and trauma consultants (Figure 3).

We applied visual management principles using the 5S tool (Table 1) [31] to develop a template that ensured entry fields were only gathering information necessary to complete triage. The triaging consultant was prompted to document three key fields:The urgency of the appointment is classified by the surgeons at the study site as operative routine, non-operative routine, and non-operative urgent.Which trauma consultant was the patient referred to by the triaging consultant.Whether or not a physiotherapy referral was advised.

The tracker template and other solutions (1–5) were implemented as a pilot before a repeat audit of the process. The audit permitted further analysis of the appropriateness of some template fields. Departmental resources were the same pre- and post-intervention.

## 3. Results

Pre-intervention patient record data on wait times were recorded on MEDITECH over three months between October and December 2021 and post-intervention in March 2022. This time period was used to analyze each of the three wait times and the number of surgeries. 

### 3.1. Reduction in Patient Wait Times

The primary goal was achieved with more than 50% of patients (52%) attending the trauma orthopedic consultant within 10 days of their ED visit. This was an increase of 11% in baseline outcomes (Figure 4). The data illustrate a reduction in the total mean wait time between an ED visit and a trauma orthopedic appointment from 17.6 (±20.1) to 11.6 (±20.1) days. The improvement represents a 34% reduction (Figure 4) and was achieved by addressing the waiting time NVA identified in the TIMWOODS. The project also improved outcomes relative to the hospital’s KPIs.

### 3.2. Reduction in Registering and Triage time, Touch Points, and IT System

Gemba interventions, pre- (*n* = 10; December 2021) and post- (*n* = 10; August 2022), were completed on time and identified touchpoints in the process. An updated post-intervention TIMWOODS (Table 3) illustrates the impact of piloted solutions (numbers 1, 2, 4, and 5). The table details a reduction in the time taken to process a referral by 52%. Touchpoints are also reduced by 51% for patient services staff and 27% for consultants to triage referrals. The table also highlights other improvements to NVA, such as re-work due to inadequate information (Overprocessing, Overproduction, and Defects). 

### 3.3. Increased Trauma Consultant Clinic Capacity and Surgical Cases

Consultant (*n* = 6) clinic caseload was analyzed pre- (January–February 2022) and post- (April–May 2022) (Table 4). The solutions (numbers 1, 2, 4, and 5) were implemented in March 2022 for a trial month. The six trauma orthopedic consultants reviewed 51% more patients post-intervention. This indicated an improved capacity for consultants to review their post-operative caseload. A 22% increase in clinic capacity was also noted. Of the patients referred from the ED to trauma orthopedics, nearly one-quarter required surgery (22%) (Table 4). This was an improvement of 8% and indicated that with these new solutions, the consultants were assessing a more appropriate caseload. The improvement assisted in addressing the skills waste identified in the TIMWOODS analysis. 

### 3.4. Patient and Staff Satisfaction

In total, pre- and post-intervention VOC was completed with consultants (*n* = 5), other staff (*n* = 9), and patients (*n* = 40). The post-intervention survey of patients indicated that both trauma orthopedic center key performance indicators (KPI) developed by the hospital were relevant and important in their care. The first KPI asserts that patients should receive a text from the orthopedic center confirming receipt of the referral within 72 h of their ED visit. The second KPI states that patients should receive an orthopedic appointment within 10 days of their ED visit. On average, patients believed that an appointment within 10.77 days (range 0–28 days) was necessary, with some stating they would consider seeking care elsewhere after this. If advised by the triage consultant, 70% of patients would be interested in pursuing physiotherapy as an alternative initial option. As the triage system was piloted in template form, following the pilot, we engaged with the stakeholders to record their opinions on the proposed solutions (numbers 1–5) before moving beyond the pilot phase. The consultant and patient services group were very positive and eager to proceed with the proposed changes. Table 5 displays the VOC (Patient and Patient Services Team) responses to sample questions surveyed pre- and post-intervention. Solution number 4 was developed with the pre-intervention VOC in mind to manage patient expectations. The stakeholder group designed a plan to implement the ‘Patient Services Homework Tracker’ for the trauma orthopedic service. This plan included the implementation of the triage template for the service while the project team liaised with IT to finalize the development of the electronic solution. A control plan was devised to support and monitor continued improvements. The impact of the pilot solutions was monitored through stakeholder feedback and compliance with the timed KPIs.

## 4. Discussion

The original goal of the study was to increase the number of patients attending their trauma orthopedic appointment within 10 days of their ED visit. We hypothesized that this could be achieved by reducing NVA in the registration, triage, and booking processes. We slightly exceeded our initial target of 50%, offering 52% of patients an appointment within 10 days (Figure 4). The total wait time from ED to consultant appointment was reduced by 34%, with a mean wait slightly longer than our KPI target of 10 days (11.6 ± 8.4 days). Significant reductions in NVA were generated in the registration process, with a 51% decrease in touch points required to register a patient (Table 3). This reduction in administrative workload corresponded with an improvement in staff satisfaction with the process (Table 5).

Consultants also reviewed more appropriate patients following the implementation of the solutions, with 22% of patients referred from the ED to the trauma orthopedic consultant requiring surgery. This indicated that consultant surgeons were assessing more appropriate patients. Although this was not specifically mentioned in patient surveys, those receiving care in the intervention phase reported a slightly improved satisfaction score relative to the baseline measures. The improved efficiency of the triage process, with patients seeing an appropriate medical professional at the right time, coincided with consultants having a greater opportunity to review their elective caseload. An elective patient signifies the non-trauma cohort of a consultant’s caseload, and these clinic appointments increased by 51% during the post-intervention phase (Table 4). This improvement was achieved through enhanced information gathering in the triage template and the option for consultants to refer patients not requiring surgical input to physiotherapy. This therapy pathway has been demonstrated successfully elsewhere, with physiotherapists participating in the triage process [9] and reviewing trauma orthopedic patients post-fracture [34].

The success of the project was enabled by extremely motivated internal champions and substantial senior-level management support, both of which have been shown to be hugely beneficial when completing an LSS project [35]. Senior management backing is a key factor in the success of LSS implementation, as a perceived lack of support can reduce project members’ motivation and result in project failure. Leaders have the potential to be role models for change and facilitators of LSS implementation [36]. The noted successes of previous Lean champions in supporting LSS initiatives are thought to have contributed to the creation of an improvement culture within the organization [37]. The fundamental desire of healthcare providers across each specialty to improve the service for trauma patients was noteworthy in this project. The principles underpinning the project were observed to align with participants’ core values of person-centered care [12]. Waring and Bishop [38] argue that LSS initiatives succeed through the mediation of the people involved as they interact and engage within enabling social structures. The organization continues to encourage staff to identify and engage in improvement projects by ensuring that they have access to education, training, and development opportunities that provide the tools and techniques to implement change. Additionally, these opportunities shape their attitudes and practices so that practice and process improvements become embedded and sustainable. 

The solutions implemented in this study will positively affect the orthopedic center and improve patient flow through the department. However, later in the patient journey, due to the change in referral options, we recognized that there was a limitation: the potential for increased pressure on physiotherapy resources. Similarly, an increase in cases requiring surgical management will test an already busy Operating Room Department. This potential shift in workload from orthopedics to other departments would not represent respect for colleagues, a key principle of person-centered care and Lean [12]. Nor would it represent systems thinking, another key principle of Lean. To manage any potential risk of creating an uneven workload across the process, department managers, frontline staff, and local LSS practitioners in each department agreed to monitor referral numbers and clinic availability on a monthly basis. At a higher level, senior management reviews seasonality patterns in service, specialty staffing levels, and bottlenecks in the system. In the event of uneven workload or a department potentially becoming overburdened, the process will be revisited to identify if any NVA exists using, for example, Gemba or the TIMWOODS tool. The project team intends to continually work with stakeholders to develop a service that is person-centered throughout the organization.

A person-centered approach to Lean Six Sigma acknowledges the imperative of developing positive relationships between service users and staff to improve the healthcare experience for all [12]. It enabled the team to understand that a shift of workload from one department to another without agreed contingency planning would fail to consider the potential negative impact on other parts of the organization and colleagues’ workloads. Oshry [39] highlights the importance of awareness of systems and an appreciation of the conditions under which colleagues work in different parts of a system. In recognition of this, the project team comprised a mix of staff levels across various administrative and clinical roles. The diversity of perspectives aimed to limit the influence of personal, positional, professional, or disciplinary bias on diagnosis or solution formation [12] and to create the conditions to enable staff empowerment and engagement [40,41]. The involvement of frontline staff in the co-creation of solutions has been shown to contribute to the development of a sense of ownership in the system and improve the prospect of whole-system success [41,42,43].

The data from this project will be used to track the number of referrals and consequent workload on hospital services involved in the new pathway. Links between each department and clinical group have already been developed to ensure timely communication. These channels include email updates to consultants and physiotherapists regarding current clinic availability. ED and orthopedic patient services meetings are scheduled along with monthly trauma orthopedic meetings. A potential benefit across the wider hospital includes the opportunity for other outpatient departments to have patients referred, registered, and triaged using the ‘Patient Services Homework Tracker’. This system will also ensure triaging decisions are documented in patients’ medical charts, further enhancing communication using the ISBAR tool (Identify, Situation, Background, Assessment, Recommendation) [44].

We acknowledge that the study was completed at a single ED site. High trauma and complex trauma presentations were not triaged to this center. However, complex cases are very unlikely to enter the outpatient service through a triage system. The study results relied on a prototype rather than the finalized Tracker system. This phase depended heavily on the support of IT, which had a number of other simultaneously occurring projects. To take account of this, improvements to the process, separate from software implementation, were made. These included the provision of educational materials, a review of triage rosters, and the referral option to physiotherapy, all of which may be transferable to other sites. Further improvements, measurements, and analyses will be completed once the tracker is finalized. Gaining a deeper understanding of the service with the tracker in situ is the initial aim. However, future projects may consider the potential for injury-specific ED to trauma orthopedic pathways.

## 5. Conclusions

The application of LSS methodology to the process of registering, triaging, and booking ED patients for orthopedic appointments resulted in a reduction in wait time for this cohort. Further reductions in NVA were noted in time taken and touch points to complete administrative tasks, as well as improvements in consultant clinic efficiency. The use of LSS principles allowed the team to quantify the existing process, identify areas of waste, and thereby deliver more effective solutions. The project has resulted in long-term changes with improved customer satisfaction and a system to triage patients for the trauma orthopedic pathway, which has the potential to be rolled out across other outpatient specialties in the hospital. In addition, changes to the orthopedic triage pathway will allow patients to access physiotherapy and provide a means for the department to sustain improvements in the efficiency of consultants’ caseload management. The project also reflects the significant contribution and importance of strong senior management support and a multidisciplinary project team.

## Figures and Tables

**Figure 1 healthcare-11-02626-f001:**
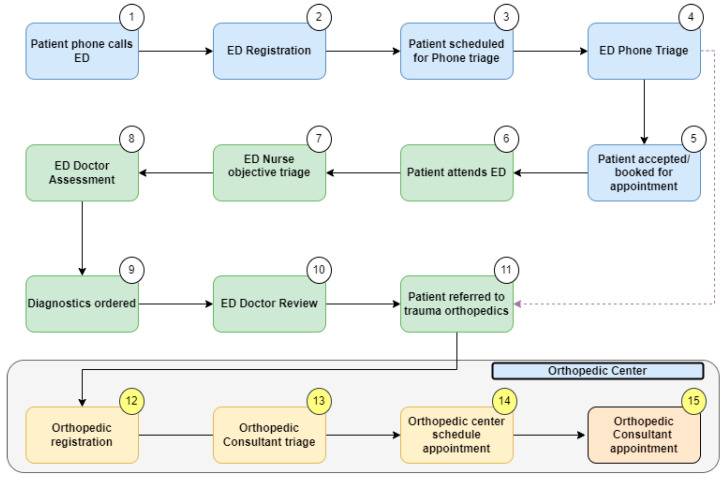
High-level process map.

**Figure 2 healthcare-11-02626-f002:**
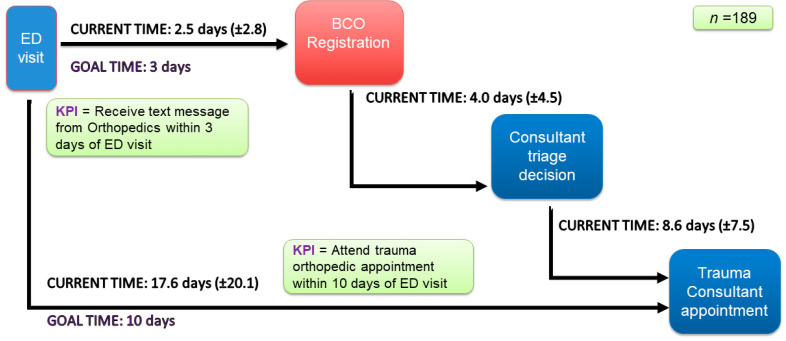
Linear process map: pre-intervention waiting times.

**Figure 3 healthcare-11-02626-f003:**
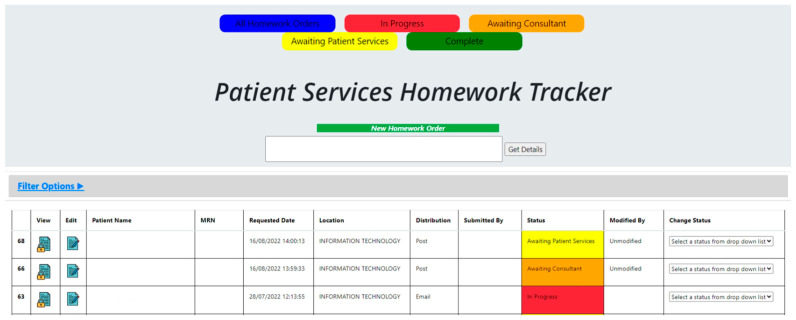
New Tracker Prototype.

**Figure 4 healthcare-11-02626-f004:**
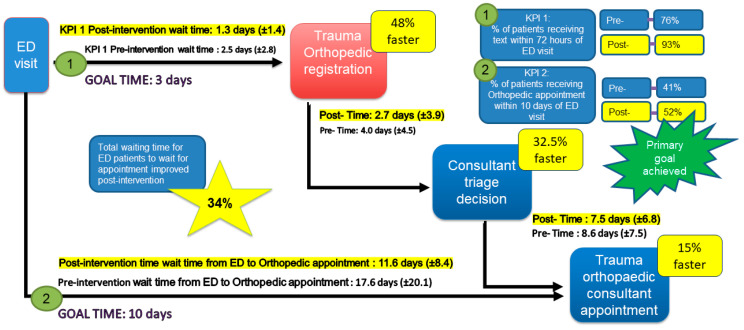
The impact on waiting times.

**Table 1 healthcare-11-02626-t001:** Lean Six Sigma tools are used by the project team.

Improvement Tool	Description
Project Charter [22]	A project charter defines the problem statement and establishes baseline data for the project. It is used to identify goals and the scope of the project.
SMART goals [23]	SMART is used to manage the project goals to determine if they are Specific, Measurable, Achievable, Relevant, and Timebound (SMART).
SIPOC [24]	A high-level SIPOC (Supplier, Input, Process, Output, and Customer) highlights the process steps and defines the customers and stakeholders.
RACI [24]	RACI (Responsible, Accountable, Consulted, Informed) identifies which stakeholders are responsible and accountable and which need to be kept informed or consulted.
CTQ [25]	The CTQ (Critical to Quality) is designed to capture the key measurable characteristics of a process or service whose performance standards must be met to satisfy the service user.
VOC [26]	The Voice of Customer (VOC) tool engages with the customer to gather their feedback about their experiences with and expectations for your products or services.
Gemba [27]	Observation of the actual process taking place
Fishbone [28]	Identifies root causes, representing the effect and the factors or causes influencing it.
FMEA [28]	Failure Mode and Effect Analysis is a risk analysis tool that is used to prevent an event from happening. It highlights the aspects of a process that should be targeted for improvement.
Process Map [29]	Process mapping supports a better understanding of complex systems and the adaptation of improvement interventions in their local context.
TIMWOODS [30]	A useful tool wherein each letter stands for one of eight potential wastes: Transport, Inventory, Motion, Waiting time, Overprocessing, Overproduction, Defects, and Skills (TIMWOODS)
5S [31]	A system to reduce waste and optimize productivity through maintaining an orderly workplace and using visual cues. A cyclical methodology of ‘Sort, Set in Order, Shine, Standardize, and Sustain the cycle.
PICK Chart [32]	Possible, Implement, Challenge, Killed (PICK) chart is a visual tool to prioritize the potential improvements to give the biggest reward.
Practicality Tool [30]	Ranks solutions in terms of the way out there, quite impractical, might be workable, close workable, and could be implemented today.
Control Plan [33]	What was measured, why, who is responsible, and what actions are required.

**Table 2 healthcare-11-02626-t002:** TIMWOOD tools are used to categorize waste in the process.

	Waste	Impact	Identified
T	Transport	Moving information	- Excessive time taken to process referral - Excessive change between software to process referrals
I	Inventory	Information unavailable	- Referral details are in different formats and are difficult to locate - Various methods of referrals
M	Motion	Excessive movement in workspace	- Numerous touch points to process referrals - Numerous touch points to triage referrals
W	Waiting time	Waiting for information or items to arrive	- Staff searching for correct patient details - Staff searching for referral information - Staff waiting for triage outcome - Patient waiting for an appointment
O	Over- processing	Doing more work than is necessary	- Logging the same information repeatedly in different software - Re-checking previous patients for updates - Re-work contacting other teams due to an incomplete referral
O	Over- production	Doing work before it is needed	- Duplicate entries on Meditech and Excel
D	Defects	Mistakes or errors requiring re-work	- No diagnostics are available - No referral available; patient calling department - Patient Services team did not email the referral - Patients attending trauma orthopedic consultants unnecessarily
S	Skills	Not using workers for their abilities	- Not utilizing new patient services staff for administrative tasks - Not utilizing physiotherapy for assessment or triage - Not utilizing trauma orthopedic consultants for appropriate patient caseload

**Table 3 healthcare-11-02626-t003:** TIMWOOD tools illustrate the intervention impact.

	Pre-Intervention	Post-Intervention
T	Excessive time taken to process referrals Excessive change between software to process referrals	Time taken was reduced by 52% Changes between software reduced from 17 to 7
I	Referral details in are different formats and are difficult to locate Various methods of referral	Referrals from ED received via tracker only
M	Numerous touch points to process referrals Numerous touch points to triage referral	Touch points were reduced by 51% for patient service staff Touch points were reduced by 27% for consultants to triage referrals
W	Staff searching for correct patient details Staff searching for referral information Staff waiting for the triage outcome Patient waiting for an appointment	Overall waiting was reduced by 34% Reduced searching for documents as all are accessible via the tracker
O	Logging the same information repeatedly in different software Re-checking previous patients for updates Re-work contacting other teams due to an incomplete referral	Excel will not be used; the tracker is a live document. Only completed referrals will be forwarded to Orthopedics via the tracker
O	Duplicate entries on Meditech and Excel and review past fields	Tracker is a live document categorizing referral progress through ‘Awaiting consultant’, ‘Awaiting patient services’, ‘Complete’, etc.
D	No diagnostics are available No referral available—patient calling department Patient Services team did not email the referral Patients attending trauma orthopedic consultants unnecessarily	Transparent and up-to-date viewing of referrals possible for ED and orthopedics using tacker Consultant has the option to triage to physiotherapy if appropriate
S	Not utilizing new patient services staff for administrative tasks Not utilizing physiotherapy for assessment or triage Not utilizing trauma orthopedic consultants for appropriate patient caseload	More efficient triage process will release time for patient services staff to complete other duties Orthopedic and physiotherapy staff are assessing appropriate patients due to the triage structure

**Table 4 healthcare-11-02626-t004:** The impact on consultant caseload.

Consultant Caseload (*n* = 6)	Pre-Intervention Number	Post-Intervention Number	% Change (+/−)
Elective new appointment	257	245	−4.6
Elective review appointment	361	546	+51
Trauma new appointment	85	97	+14
Total appointment number	702	878	+22
**Surgery Post-ED Referral**	**Pre-Intervention** **%**	**Post-Intervention %**	**% Change**
Trauma surgery	6.3	12	+5.7
Non-trauma surgery	5.2	7.5	+2.3
Total surgery	14	22	+8

**Table 5 healthcare-11-02626-t005:** The impact of solutions on Voice of the Customer (VOC).

Patient Survey	Pre-Intervention	Post-Intervention
What would be your expectations regarding the wait time between your ED visit and the consultant appointment?	10.77 days (range 0–28 days)	10.3 days (range 0–28 days)
What is your satisfaction level based on your experience with the current process?	6.1/10	6.6/10
Would you avail of an Allied Therapy appointment if advised by your triaging Orthopedic consultant?	70% see Allied Health	80% see Allied Health
Orthopedic patient services survey	Pre-intervention	Post-intervention
What percentage of your working week is spent registering ED patients?	30%	30%
Approximately how long does it take to register a trauma referral from the ED?	10 min	5 min
How satisfied are you with the current process?	2 (Low)	4 (High)

## Data Availability

All data are represented within the paper.

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
