# Peer review of "Using Lean Six Sigma in a Private Hospital Setting to Reduce Trauma Orthopedic Patient Waiting Times and Associated Administrative and Consultant Caseload"

_healthcare, 2023, doi:10.3390/healthcare11192626_

Round 1

Reviewer 1 Report

This is an interesting manuscript.  

Line 104 Include only ED patients referred for peripheral musculoskeletal symptoms.. It would have been valuable to have some examples of the type of injuries the patients had because some required surgeries and others physiotherapy. In line 105 one can see those who were excluded. 

Line 184 n=30 purposive sample was surveyed by phone call. How many were the patients in total? How did you choose those 30 patients? 

3. Results Line 267 the primary goal was achieved with more than 50% of patients (52%). But many were they? Was it something specific that they had in common?  

3.3. Increased trauma consultant. Line 294 Consultant (n=6) clinic caseload was analysed. Did every orthopedic consultants available participate? 

3.4. Patient and staff satisfaction 

Line 308. How did you choose those who participated in this survey? In table 5 are questions and responses from the survey in line 330. Were these the only questions? Were there any background questions? 

I also wondered why 10 days (about 1 and a half weeks) were the limits? Was it based on something from literature? 

The authors described their goals and hypothesis at the end of the introduction. They used “satisfaction” in their hypothesis, but I could not see that word used in their discussion or conclusion. I wondered if that is the right word. The project was effective, and they reached their goal of reducing the waiting time but it would have been even better to see more information about the participants.

Author Response

Dear Reviewer, thank you for your comments, please see the attachment.

Reviewer 2 Report

Considering the constantly increasing work-load effective strategies to take appropriate care of all patients are required. The authors conducted an interesting study and their approach was shown to improve patient care in terms of total wait time and reducing unnecessary steps via slight modifications that can be easily implemented. Altogether I’d like to congratulate the authors on this study. A few minor revisions are required as well as a ethical vote statement.

P2 line 45 – rephrase – e.g. was carried out etc.

P2 line 55ff – simplify and rephrase

Please provide a description of LSS 

Ethical Vote???

Minor edits required, otherwise fine

Author Response

(The authors gave the same response as above.)

Reviewer 3 Report

The research aims to address the challenge of long waiting times for orthopedic patients in a hospital in Ireland. Using Quality Management principles such as Lean Six Sigma and Define, Measure, Analyze, Improve, and Control (DMAIC), the authors present a case of comparative study with pre-post intervention and report that the total wait time of patients for review being reduced by 34%, a 51% reduction in process steps required for registering and an increase in orthopedic consultant clinic capacity of 22%. Overall, it is a well-designed pilot study with interesting findings that can motivate other hospitals to consider adopting Quality management principles for improving patient experience. However, I have some concerns and suggestions that I would like the authors to consider/address to improve the quality of the manuscript.

·         I am a bit confused. Are all patients accessing the orthopedic center through the ED? If not, was there a change in the volume accessing the orthopedic center?

·         Are all the improvements in Table 3 and Table 4 attributable to the tracker? Was that the only change? Or could there be any confounding factors?

·         Also, in the abstract, the authors say, “increase in orthopedic consultant clinic capacity of 22%.” Does that mean in the past, the physicians had open slots? Is there a metric such as utilization that can be used?

·         Are the number of beds, physician slots, and other resources the same during the pre and post-intervention study?

·         I noticed that the Goal time is ten days, but currently, it is 11.6 days. Are the authors working on other interventions to improve that?

·         Also, on the same note, the authors mention that the tracker was Solution 1, the one which can have the most impact. Can the authors please add other solutions as well? 

No major comments. 

Author Response

(The authors gave the same response as above.)
